# Variation in spatial dependencies across the cortical mantle discriminates the functional behaviour of primary and association cortex

Robert Leech [1] ✉, Reinder Vos De Wael[2], František Váša [1], Ting Xu [3], R. Austin Benn[4], Robert Scholz [5], Rodrigo M. Braga[6], Michael P. Milham[3], Jessica Royer[2], Boris C. Bernhardt [2], Emily J. H. Jones[7], Elizabeth Jefferies[8], Daniel S. Margulies[4] & Jonathan Smallwood [9]

Recent theories of cortical organisation suggest features of function emerge from the spatial arrangement of brain regions. For example, association cortex is located furthest from systems involved in action and perception. Association cortex is also 'interdigitated' with adjacent regions having different patterns of functional connectivity. It is assumed that topographic properties, such as distance between regions, constrains their functions, however, we lack a formal description of how this occurs. Here we use variograms, a quantification of spatial autocorrelation, to profile how function changes with the distance between cortical regions. We find function changes with distance more gradually within sensory-motor cortex than association cortex. Importantly, systems within the same type of cortex (e.g., fronto-parietal and default mode networks) have similar profiles. Primary and association cortex, therefore, are differentiated by how function changes over space, emphasising the value of topographical features of a region when estimating its contribution to cognition and behaviour.

One of the most important discoveries in human neuroscience is that brain topography plays an important role in determining how a region contributes to cognition and behaviour[1]. These topographic features can shape a region's function in many ways including: (i) through the influence of neighbouring neural systems that make up the local environment within which a specific region is embedded[2], (ii) the physical location of the network on the cortical mantle with respect to core cortical landmarks[3], (iii) and more abstract topographical features such as the degree to which functional activity within a network is spatially distributed across the cortical mantle[2,4], or, instead is limited to adjacent regions, often within a single cortical lobe[5,6].

Contemporary evidence suggests that local topographical properties influence a region's function in a complicated, interdependent manner. For example, neural systems concerned with sensation and movement, such as the visual or motor cortex, are spatially distant from each other, yet both of these systems tend to be relatively spatially contiguous, and both contain topographic features resembling maps, either of the external environment or how the organism engages with the outside world[7-10]. Other systems, such as the default mode or frontoparietal networks, are located in regions of association cortex, are spatially adjacent to one another, both are spatially distributed across cortex; yet functionally these

[1]Centre for Neuroimaging Science, King's College London, London, UK. [2]McConnell Brain Imaging Centre, McGill University, Montreal, QC, Canada. [3]Center for the Developing Brain, Child Mind Institute, New York, USA. [4]Integrative Neuroscience and Cognition Center (UMR 8002), Centre National de la Recherche Scientifique (CNRS) and Université de Paris, Paris, France. [5]Max Planck School of Cognition, Leipzig, Germany. [6]Neurology, Interdepartmental Neuroscience Program, Northwestern University, Evanston, IL, USA. [7]Centre for Brain and Cognitive Development, Birkbeck College, University of London, London, UK. [8]Department of Psychology, University of York, York, UK. [9]Department of Psychology, Queens University, Kingston, ON, Canada. ✉e-mail: robert.leech@kcl.ac.uk

systems appear to serve different, often opposing roles in human cognition[11]. Topography is also important for understanding macroscale brain function, because systems that tend to be more spatially discontinuous (e.g., the default mode network) tend to be more distant from sensory and motor systems where spatial discontinuity is an exception rather than the norm (e.g. sensorimotor or visual cortex)[3]. In contemporary neuroscience, macroscale topographical features provide a useful heuristic for understanding the involvement of frontoparietal and default mode networks in cognition. These networks are hypothesised to be at the transmodal apex regions of a broad sensory-fugal hierarchy, allowing oversight across broad areas of cortex[12]. In contrast, mesoscale features of topography, such as the retinotopic maps located within sensory cortex, are thought to explain aspects of how the visual system represents and extracts features of the environment from retinal input[8].

Topography at both macro and mesoscale is, therefore, a key principle of brain organisation and is crucial for understanding brain function both within specific systems and across the cortex as a whole. Our study set out to formally examine how the meso and macro scale perspectives can be combined to formally understand the relationship between topography and brain function. The distance between regions, calculated as the geodesic distance between two vertices, provides one metric to understand how topography influences function. This measure has been used to describe macroscale features of cortical topography, for example, highlighting that systems like the default mode and frontal-parietal cortex are distant from both systems concerned with sensory input and motor output systems[13]. However, a given location on the cortical mantle may be influenced by local topographical features as well, such as the features of the local neighbourhood in which the region is situated, or, whether the system is part of a distributed or localised network. Accordingly, it is important to understand how the balance of meso and macro scale influences combine in order to understand how topography influences function within a given brain region. Our study set out to understand meso and macro scale changes in the influence of topographical features on brain function by examining whether there are regional differences in the way distance impacts functional connectivity.

In order to establish how distance between regions influences their similarity in function, we calculated for each cortical surface vertex how the similarity of its activity changes with all other vertices as a function of the distance between them; quantifying the local rate of change of similarity across the cortex. This is a simplified version of the empirical variogram[14], as illustrated schematically in the upper panel of Fig. 1. Spatial variograms are expected to show that similarity in function declines with distance until it reaches an asymptote, the distance after which there is no longer a spatial dependency between vertices. The empirical variogram can be summarised by fitting an exponential function which in turn can be described by two values capturing how similarity changes with distance for each vertex: the effective range and the sill. The sill is the height (i.e., degree of dissimilarity between two regions) and the range is asymptote (i.e., the spatial distance between the two regions). Heterogeneity in the spatial variogram across regions can be used to quantify the different ways topography influences function in different cortical locations. For example, in regions where function is more influenced by the local neighbourhood, the spatial variogram shows a relatively shallow decline in similarity with distance. In contrast, in regions where function is relatively distinct from the local environment, the variogram should increase more rapidly with the distance. This approach allows for the presence of variable spatial dependencies across the cortex, in contrast to accounts that imply a homogeneous spatial relationship, e.g., a single exponential distance rule[15].

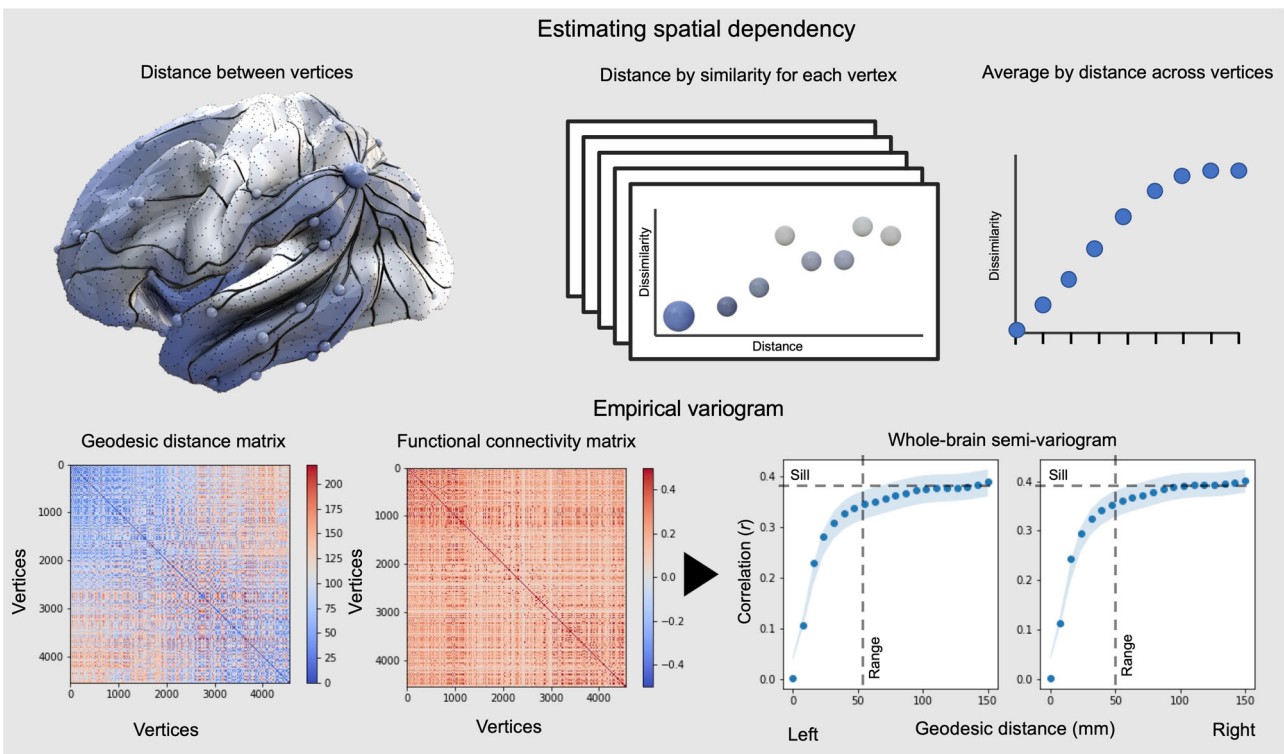

**Fig. 1 | Calculation of variograms.** Top Panel. Schematic illustration of how spatial variograms can be used to characterise how functional connectivity changes as distance increases between brain regions. Bottom Left. Whole-brain variograms of functional connectivity can be calculated by comparing how the distance along the cortical surface is related to the average similarity in brain activity between regions. Bottom Right. Whole brain variograms are shown for the left and right cortices and can be seen to be broadly similar. The thick lines/dots are the mean across participants, and the filled area is the standard error of the mean. The dashed lines are the estimated location of the sill (asymptotic correlation between vertices) and range (distance in mm between vertices at which the asymptote is reached).

## Results

### Whole-brain spatial dependency

We first quantified the spatial dependency between functional connectivity and distance by calculating whole-brain variograms assessing how functional connectivity (Pearson's correlation) varies with distance along the cortical surface for each hemisphere (Fig. 1, lower panel). We used resting state fMRI data from 51 participants from the Human Connectome Project. We took two scans on the same data for each individual allowing us to calculate the reliability of these metrics within an individual. Averaging these vertex-wise variograms across the whole cortex, the global variogram, reveals an initially steep rise (rapidly increasing dissimilarity with distance (Fig. 1). This is followed by a continuous increase up to the measured limit (all vertices included distances up to 150 mm, which was the maximum distance present for all vertices (see Supplementary Figure 1 for vertex distance distributions and higher upper measurement limits). The variograms for the left and right hemispheres show a similar pattern (see left hand panel). The landscape of these variograms can be formally understood by comparing the observed rate of change in function with distance with different mathematical growth functions (e.g., exponential, gaussian, sinusoidal and power-law). It can be seen in Fig. 1 that the whole brain variogram of the human is most similar to an exponential relationship.

For the purposes of our analyses, we extracted the two parameters used to fit the theoretical function to the empirical variograms: (i) the sill, which is the height the variogram reaches at 95% of its asymptote and reflects the approximate point at which there is no longer a relationship between space and functional connectivity (i.e., that vertex' baseline average correlation level with other vertices); and (ii) the range which is where the sill occurs. These are both displayed in the top panel of Fig. 2. Importantly comparing the variogram calculated for each of the participants from separate resting state scans on the same day shows a high degree of correspondence both in terms of the sill (the average difference in correlation between vertices) and the distance (i.e., rho > 0.73; Fig. 2 top panel).

### Regional variation in spatial dependency across the cortex

The whole-brain variograms establish that in humans, distance leads to an increase in dissimilarity in neural function that is asymptotic exponential in nature and that these measurements are broadly consistent within an individual over time. This aligns with descriptions of spatial similarity previously reported in humans and non-human primates e.g., refs. [15–18]. By computing variograms, we are able to go beyond a single description of spatial dependency in each region of the brain, and this therefore allows us to capture regional differences in spatial dependencies (see also Supplementary Fig. 2, for random models with homogeneous spatial dependency structures to contrast with the empirical results). To understand whether there are systematic differences in how distance leads to changes in neural function across different brain regions, we calculated separate variograms for each vertex across the cortex. The middle panel in Fig. 2 summarises how the two metrics (sill and effective distance) vary across the cortex. It can be seen that sill (reflecting the spatial dissimilarity in functional connectivity across the cortex) ranges between 0.25 and 0.5, and that in some regions the dissimilarity continues to increase to the maximum range of our measurements (150 mm).

### Relationship between spatial dependency and cortical organisation

Having highlighted the features that whole brain variograms have, we next considered how this varied across the cortex. To this end, we examined how the distribution of the sill and the effective range varies across the principal gradient of change in functional connectivity[3]. This gradient can be derived by application of dimensionality reduction techniques to functional connectivity data[3], and recapitulates foundational features of the sensory-transmodal cortical hierarchy[1].

The lower panel of Fig. 2 shows that regions closer to the transmodal end of the principal gradient tend to be regions where the variograms tend to have a relatively high sill and short effective distance (i.e., regions where dissimilarity shows a relatively rapid increase), in general. In contrast, regions closer to the unimodal end of the principal gradient tend to have a relatively lower sill and a longer effective distance (i.e., regions that show a slower rate of decline in function with increasing distance). This analysis provides preliminary support that two broad types of cortex (primary and association cortex) can be discriminated based on how activity varies with distance. Spin permutation tests (Fig. 2, bottom, right) as well as generative null models based on randomisation or randomisation followed by smoothing with a homogeneous function (Supplementary fig. 2) show that these relationships are unlikely to be due to chance.

The principal gradient provides an organising principle for macroscale features of brain function, including large-scale brain networks (see ref. [3]). Next, we examined how the large-scale networks that span the principal gradient, focusing on a well-defined set of canonical resting state networks from Yeo and colleagues[4]. Figure 3 (upper panel) shows the average empirical variogram for each network while the lower panel shows the average sill and effective distance of each network. Regions making up the limbic network (Cream) have the highest sill and the shortest effective distance, a pattern that is also seen in the transmodal networks (Default mode, Red; Fronto-parietal network, Orange) but to a lesser degree. Regions that make up unimodal cortex (Visual network, Purple; Motor cortex, Blue) show the reverse profile with variograms with small sills and relatively long effective distance. Finally, the two attention networks (Dorsal and Ventral) show intermediate profiles both having moderate sills and effective ranges. These two systems are distinguished from each other because the Dorsal attention network has a longer effective distance and a short sill, and so is more similar to the unimodal systems, whereas the ventral attention network shows the opposite profile.

This network analysis contrasts with a comparison between broad features of brain organisation such as the principal gradient. In particular, while there are clear differences between networks in terms of their variogram profile, networks embedded in similar types of cortex show relatively high similarity. In particular, both the default mode network and the frontoparietal network, embedded within association cortex, show similar profiles. Likewise, the variograms of visual and motor systems, which are both embedded in primary cortex, are also similar. To quantify this apparent similarity we randomly permuted the location of the Yeo networks (by rotating them on the sphere) to generate null models and compared the difference in the Range and Sill parameters. This analysis showed the only significant differences were between different types of cortex (see Fig. 4), e.g., between visual and default mode networks.

Having established that heterogeneity in spatial dependencies capture important features of brain organisation in humans, we next sought to understand whether this generalises to non-human primates. To this end, we repeated this analysis in a sample of macaques (using homologue networks, see Methods for details). This analysis identified that the network profile of each species is broadly similar. For example, in both species the limbic network has the highest sills and the shortest effective distances, and the visual system provides the clearest example of the opposite profile (low sills and longer effective distance). We note that some regions within the limbic network have been reported to have signal dropout and related issues in the Human Connectome Project dataset[19] and so should be interpreted with caution.

### Clustering variation in spatial dependency

The variograms stratified by resting-state network suggest that there may be a small set of spatial dependency profiles that characterise a larger number of networks, and that these likely correspond to the difference between association and primary cortex. To provide an

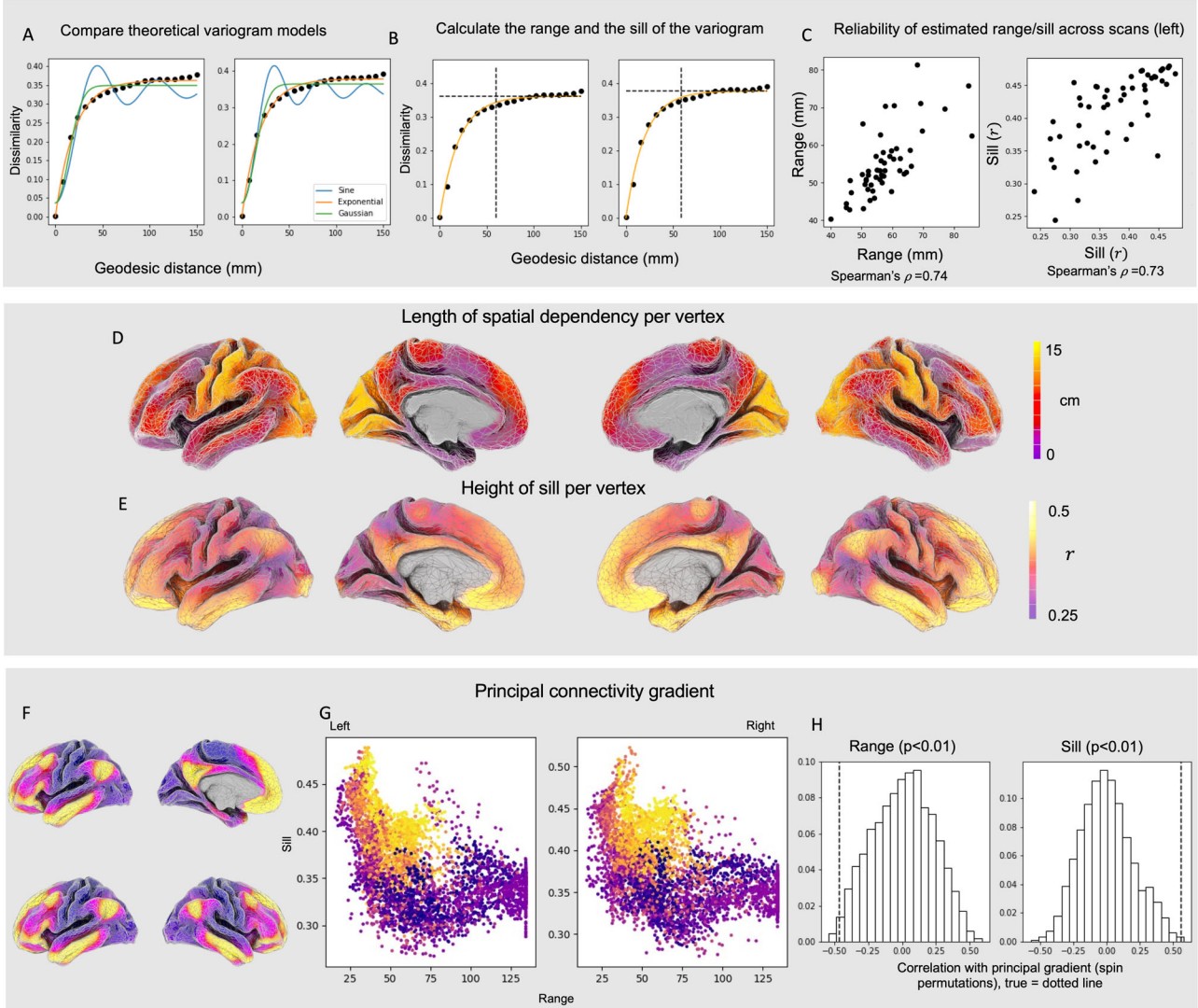

**Fig. 2 | Distribution of the sill and effective distance of variograms across the cortex. A** Variograms can be formally described through comparison of the observed rate of change between similarity in brain activity and distance with different mathematical growth functions. We observe that the whole-brain variogram has most similarity to an exponential function. **B** Variograms can be characterised by two numbers, the partial sill (the height of the curve at 95% of its asymptote) and the effective range (the distance of the sill). **C** Both the sill and the range of the whole brain variogram show reasonable similarity when measured within the same individual in two scans on the same day (>0.73). **D** The regional distribution of the range (the distance of the sill) and (**E**) the sill (the height of the

variogram at 95% of its asymptote) across the vertices of the human cortex. It can be seen that the sill varies from 0.25 and 0.5 across the cortex and that in some regions the range can be as long as 15 cm. The relationship between the **F** distribution of the principal gradient of intrinsic connectivity and (**G**) variograms at each vertex (as described by each vertex's partial sill and effective distance). **H** Spin permutation tests to assess the significance of the correlation between the principal gradient and theoretical variogram parameters (the range and the sill). The true values are depicted by the dashed lines and the histogram displays the distribution of correlations from the permuted maps.

independent test of this idea, we performed hierarchical clustering on the binned data from the vertex-wise variograms and display the results coloured by different canonical networks. The top panel of Fig. 5 presents the dendrogram produced by this analysis. Clustering vertices based on their variogram profiles gives rise to two groups, one predominantly encompassing the unimodal systems (primary sensorimotor networks as well as parts of the dorsal attention network) and the other corresponding to limbic and transmodal systems, as well as the ventral attention network. This analysis, therefore, highlights a broad dissociation of cortex into two classes based on their variograms: one class of regions where the variograms have low sills and long connectivity and a second class of regions with higher sills and shorter effective distances. We also assessed how consistent these results were for individuals' variograms across different scans (Fig. 5C), to ensure the cluster structure was not a consequence of group averaging and

generalises to out-of-sample data. Comparing each individual participant's empirical variograms across scans showed within-cluster correlations (cluster variograms from scan 1 correlated with cluster variograms from scan 2) substantially higher than across clusters.

Our analysis highlights that variograms vary between primary and association cortex, but do not separate large-scale networks such as the default mode and fronto-parietal cortex, even though these have contrasting behaviour at rest[20] and have differing functional profiles. Our next analysis, therefore, examined how the variograms vary with meta-analytic descriptions of function. To this end, we averaged vertex-wise estimates of the range and sill parameters for responsive vertices (defined as those with an estimated evoked BOLD response greater than threshold) in 24 topic maps generated by data mining the neuroimaging-related literature[21] and discovering brain maps associated with them from an automated meta-analysis[22]. Figure 5

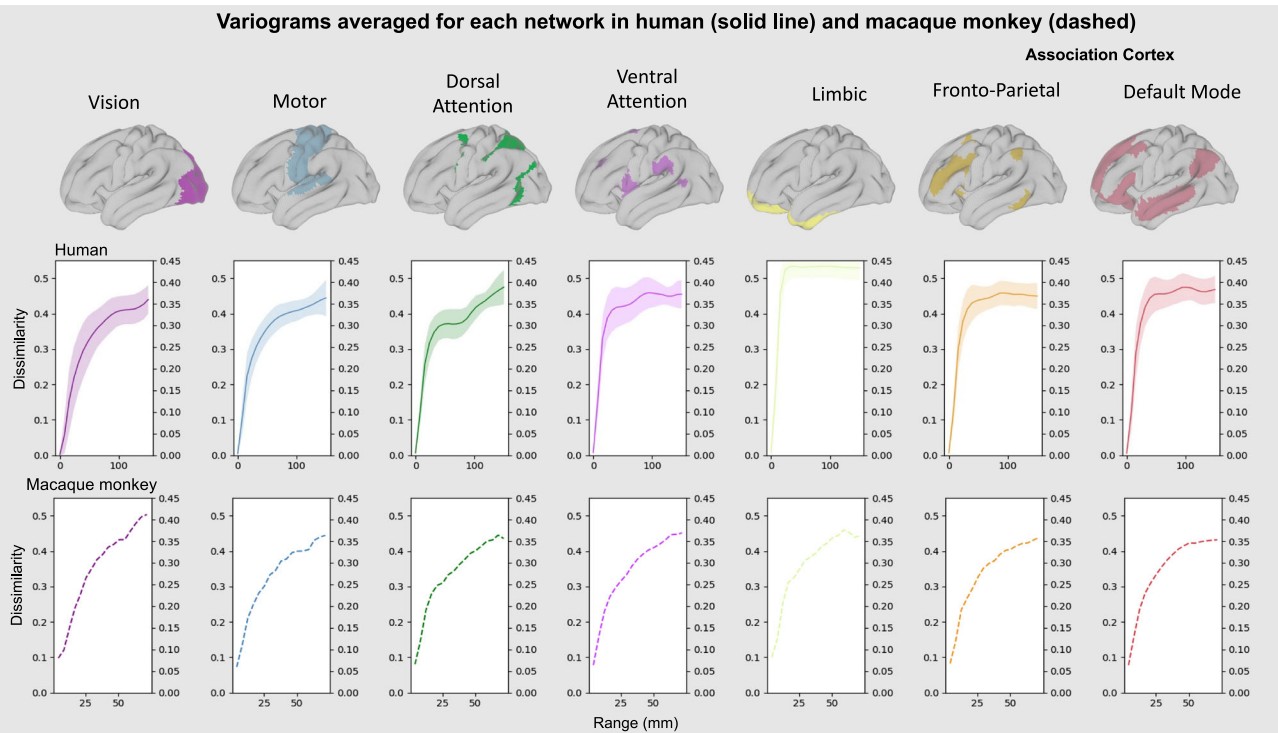

**Fig. 3 | Variograms calculated for each canonical resting state network (Yeo, Krienen et al.[47]) in humans and in homologue networks in macaques.** The middle panel shows the mean variogram (FC dissimilarity by distance along the cortex) calculated across all vertices for each Yeo network in the human Human Connectome Project data; the filled areas are the standard errors of the mean across vertices. Below is a similar analysis with fMRI data averaged from 14 awake Macaque monkey as a comparison. Data to recreate the variograms in Fig. 3 is available in source data file.

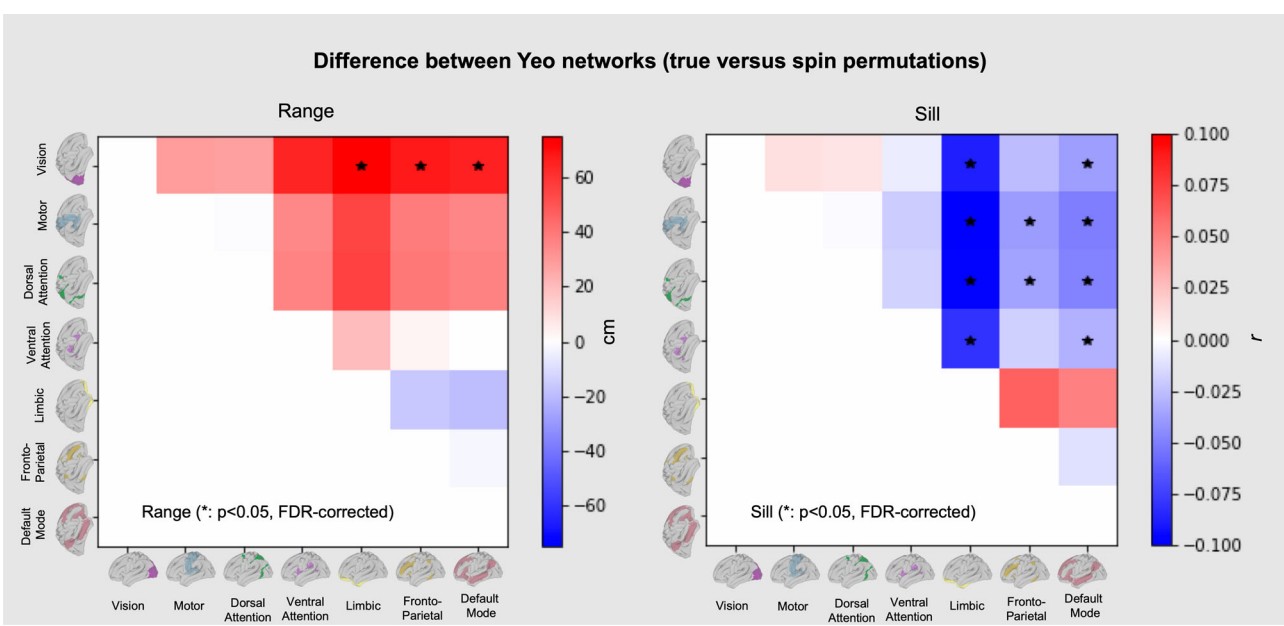

**Fig. 4 | Permutation tests to assess the difference in average Sill and Range between networks.** The results from spin permutation tests comparing the difference in the differences between the range and sill between each pair of canonical resting state networks (in the human); network pairs with significant differences (FDR-corrected, $\alpha < 0.1$) are indicated with an asterisk.

shows how brain regions related to different cognitive states differ in terms of their profile of spatial dependencies. In general, more externally focused tasks (e.g., labelled "visual" or "motor") showed slower decrease in similarity with distance and a lower sill; whereas cognitive tasks associated with more abstract functions (such as "emotion", "social", "memory"), were associated with the opposite pattern with shorter ranges and higher sills. We subsequently clustered the tasks according to their sills/ranges to allow us to easily visualise the variability in the variograms associated with each task (the red/blue colours in Fig. 5, panels A–E). This allowed us to create a composite task activation map for each cluster and plot the associated variograms showing the different spatial dependency profiles.

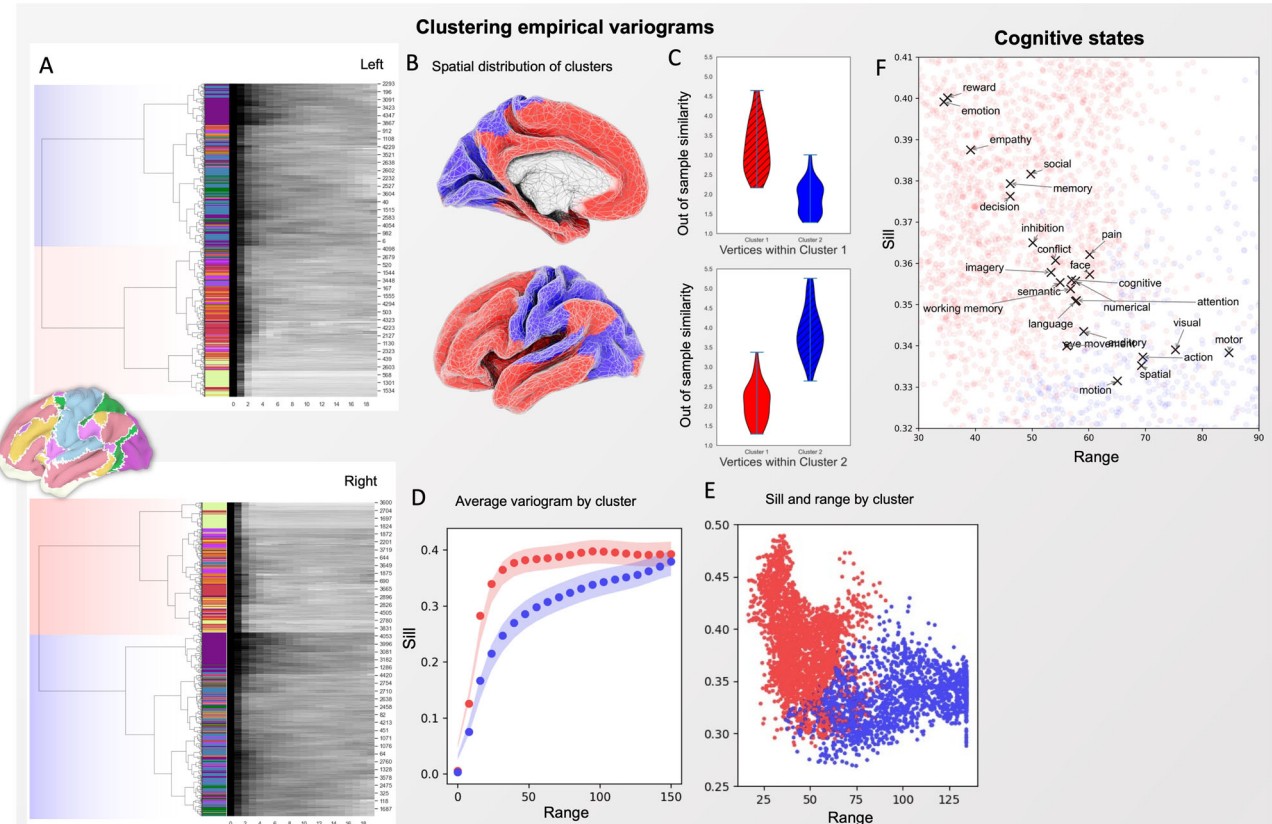

**Fig. 5 | Clustering vertices based on empirical variograms.** Left **A:** clustering vertices based on empirical variograms. The dendograms, are coloured by the Yeo network that each vertex belongs to, displaying the tree structure of the similarity between variograms; the number for each column is the index of the representative vertex. **B** The dendogram was used to cluster the data into two clusters (coloured red and blue) for the left and right hemispheres. The order of the clustering was arbitrary across hemispheres and has been coloured based on approximate similarity between the left and right hemispheres. Broadly, transmodal regions were clustered together in a separate cluster (red) to unimodal sensorimotor regions (blue). **C** Correlation of empirical variograms across vertices are consistent within each cluster within individuals and across different MR scans; bars are the standard error of the mean. **D** Average empirical variograms for each of the clusters within individuals reveals that one cluster exhibits more dramatic change in functional similarity with distance (shaded areas are the standard error of the mean). **E** The range and sill for each vertex, coloured by the cluster label for the left and right hemispheres. **F** the ranges and sills calculated across vertices activated by different cognitive processes (taken from a large automatic meta-analysis); These are overlayed on vertices coloured by their cluster membership from **E.**

## Relationship between spatial dependency and intracortical myelin

Our final analysis examined how microstructural features of different regions of the cortex correspond to the observed differences in spatial dependency profiles across cortex. Given its role in signal propagation, we examined whether myelination is linked to the shape of the variograms for different vertices. Figure 6 depicts the spatial distribution of estimated cortical myelin. We split vertices into deciles based on their levels of cortical myelination and plotted separate variograms for each decile. A clear separation emerges, with more highly myelinated vertices displaying, on average, longer distance spatial dependencies, and lower sills. This is made more explicit by plotting the range and the sill per vertex (Fig. 6) coloured by the level of myelination (warm colours indicating higher myelination).

## Discussion

Given emerging evidence of the importance of topography in the mammalian cortex[3,12], our study set out to understand how the distance between regions relates to their functional similarity. In particular, we examined whether this profile of spatial dependence is heterogeneous, varying across different cortical regions. Our analysis first established whole brain variograms are reasonably consistent across hemispheres, individuals, and within individuals measured in different scans on the same day. When we examined these on a regional basis, we observed substantial differences across the cortex.

This finding suggests a more complex relationship between functional connectivity and distance along the cortex than has typically been reported. For example, multiple previous studies have defined a homogeneous cortex- or brain-wide relationship between function and distance (such as a single exponential distance rule, e.g., refs. 15,17,18,23, although[23] noted that a single spatial relationship was inadequate to fully explain patterns of brain activity). The regional variability that we observed, reflects known functional divisions of brain function. Notably, the observed differences in spatial dependence profile recapitulated the distinction between primary sensorimotor and transmodal association cortex. In primary sensorimotor cortices, including visual and somatosensory cortex, we found that increasing distance is associated with a gradual change in function. In contrast, in association cortex we found that function changed with distance at a much faster rate. Importantly, while these broad types of cortex differed substantially in terms of their spatial dependencies, networks located within similar types of cortex were generally similar to each other, an observation which is important because these systems are often thought to have contrasting functional and cognitive associations. These differences between unimodal and association cortex in humans were broadly similar to those seen in macaques suggesting that they are conserved across the primate nervous system. We found that these changes in how distance impacts functional variation are likely to be at least partly related to differences in microstructure, as we found differences between association and unimodal

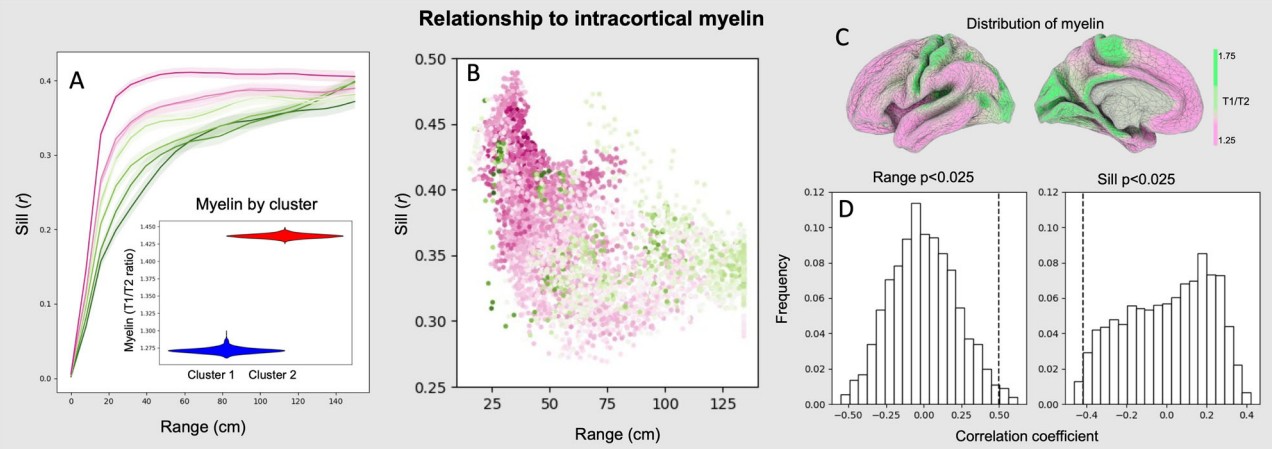

**Fig. 6 | Variograms vary with intracortical myelin. A** The empirical variograms between functional connectivity and distance split into deciles based on vertices' myelin value (pink-greener colours correspond to higher-myelin content; shaded area is the standard error of the mean across individuals). Individual average estimated intracortical myelin for the two clusters. **B** The estimated range and sill for each vertex, coloured by estimated myelin. The inset brain is the average distribution of estimated cortical myelin (from the HCP group average dataset). **C** the average estimated myelin distribution from the lateral and medial surfaces. **D** spin permutation tests comparing the spatial distribution of myelin with the range and sill parameters; the true correlations are depicted by the dashed lines.

cortex similar to those seen when exploring variation in intracortical microstructure approximated by the ratio of T1w/T2w image intensity a known proxy for intracortical myeloarchitecture[24].

These results have implications for understanding how topographic differences influence cortical function. First, our data provides support for an organisation of unimodal cortex that supports the progressive elaboration of encoded stimulus features[25]. Our analysis established that both sensorimotor cortex and visual cortex are situated within regions in which the changes in function over distance are some of the most gradual when the cortex is viewed as a whole. When contrasted with association cortex, this pattern is consistent with the view that sensory regions have a spatial organisation in which adjacent regions encode progressively complex features of the information extracted from sensory signals and that these compressed signals form the basis of signal processing for the next stage in the hierarchy e.g. ref. [26]. This pattern of progressive change is assumed to be important in regions of primary cortex, such as visual cortex, and is captured empirically by the variograms in these regions which show relatively small steady changes in functional properties as the distance between two regions increases.

Our study also provides insight into theoretical perspectives on how neural processing occurs in regions of association cortex. For example, contemporary work highlights that regions of association cortex can have relatively unique features both in terms of the functions they support, and in their observed neural properties (for a similar argument see[12]). For example, both the fronto-parietal and default mode networks are implicated in cognition in a relatively abstract manner, highlighted by their involvement in a wide range of tasks which despite being superficially different may draw on similar underlying cognitive operations. For example, situations that have superficially different features, such as the Stroop[27] or working memory[28], but show a common reliance on executive control, tend to activate the fronto-parietal network, as well as other task positive systems[29]. Similarly, the default mode network is often observed as contributing to situations when information from memory may be important for organising cognition, such as during mental time travel[30], memory processes that rely on semantic[31] or episodic knowledge[32]. Our analysis suggests that both of these large-scale systems are situated in regions of cortex where there are fairly rapid changes in functional similarity with increasing distance. These rapid changes in function over relatively short distances are likely to reflect the interdigitated nature of these systems[6,33]. These perspectives assume that a general property of associative cortex may be a topographic organisation in which relatively different functional

systems terminate within close proximity of one another. This topographic system could form the basis of an architecture that is hypothesised to explain why both the fronto-parietal[34] and default mode networks[12] contribute to multiple different forms of behaviour in a relatively abstract manner. These more complex, interdigitated patterns of function are captured empirically by the variograms which show rapid functional changes as a function of distance in each of the large-scale networks in association cortex. Importantly, our analysis suggests that both the fronto-parietal and default mode network share similar variogram profiles, suggesting that this is likely to explain similarities in their function rather than their differences.

Our study provides insights into the important observation that the default mode network, a brain system located at the maximal distance from primary landmarks like the calcarine sulcus, also has a functional profile which is one of the most unique in the mammalian nervous system[3]. Our analysis suggests regions of cortex where the default mode network is located combine two unique topographic properties that together explain why the distance between these systems and the primary sensorimotor landmarks corresponds to the primary dimension of functional differentiation with the whole brain connectivity space[3]. Our analysis suggests that the increasing distance from primary landmarks in sensory cortex, and regions of the DMN would first lead to increasing differences in functional similarity through the slow progressive changes in function with distance that emerge in primary cortex. In conjunction, with these gradual changes, our study suggests that the cortex where the DMN is where function changes most rapidly with increasing spatial distance. Thus, the observation that the distance between the DMN and sensory cortex corresponds to the greatest differentiation in function (i.e. the principle gradient of functional connectivity[3]) is inevitable because this distance combines (i) the progressive changes in function within primary sensorimotor cortex, and (ii) the complex interdigitated structure seen within the DMN[6]. Based on our analysis of T1w/T2w images it is possible that microstructural differences, such as myelin content, may be an important feature in distinguishing these types of cortex, an important question for future research to explore with more detailed anatomical techniques (e.g. ref. [35], than those used in the current investigation.

Although our study highlights how different types of cortex can be understood through the emergence of functional differentiation across space, it also raises a number of important questions for future research into how topography shapes function. First, although our study shows that association and unimodal cortex systematically vary

in how function changes across the surface of the brain, this metric does not discriminate between systems that are known to be distinctive in their functions. For example, although the variograms for both the fronto-parietal and default mode networks are similar, the situations in which these systems contribute to cognition are different. Likewise, the variograms in motor and visual cortex are similar, yet these systems have clear functional differences. It is likely that the different roles that these systems play in cognition may arise, not from the general way that function changes with space in these areas of cortex, but in terms of the specific location that these systems inhabit within the broader cortical landscape. In this way, our study highlights the more abstract properties that distinguish association and unimodal cortex, but do not provide a concrete explanation for how these systems contribute to cognition and behaviour in a distinctive manner.

Second, our study does not constrain accounts of why association and unimodal cortex have differences in the spatial differentiation that we observe. Our analysis highlights that microstructural differences, via a proxy of intracortical myelination, systematically track differences in the empirical variograms. However, there are likely to be multiple microstructural features that track these differences, and these microstructural properties may also vary as a consequence of experience. Therefore, it is important for future work to examine the different genetic and experiential changes that influence how function varies as a function of distance in both primary and association cortex to fully understand the influences that determine this fundamental feature of cortical organisation. One possibility is that the high degree of spatial heterogeneity within association cortex may result from the long-distance connections that link specific regions[36]. By extrapolation, these long-distance connections may provide a clue into how regions within these areas of cortex are able to serve distinct cognitive functions. Understanding how the broad changes in the parameters captured by the variograms relate to long-distance connections is an important question for future research to address. In addition, from a methodological perspective, it is important for future work to understand how data analysis decisions (such as smoothing) impact variation in spatial autocorrelation as well as their consequences for quantifying large-scale cortical organisation[37] and making statistical inferences.

## Methods

The research presented here complies with relevant ethical regulations (King's College London College Research Ethics Committee) governing reanalysis of existing data.

### Imaging data

The data used in this study are available from the Human Connectome Project (https://www.humanconnectome.org/study/hcp-young-adult/document/extensively-processed-fmri-data-documentation), the PRIMatE Data and Resource Exchange (https://fcon-1000.projects.nitrc.org/indi/indiPRIME.html), and Neurosynth https://neurosynth.org/analyses/topics/).

The majority of the analyses were performed on 51 participants' resting state fMRI from the Human Connectome Project's minimally pre-processed dataset (34 female); this involved registration to a common MNI152 template, minimal spatial smoothing and extensive filtering for slow drifts, motion and other nuisance signals estimated using independent components analysis[38]. The 4D fMRI datasets for each participant were projected onto the Conte32k surface and the number of faces reduced resulting in 10,000 remaining vertices (using Matlab's reducepatch command). Two resting-state runs (with opposite phase encoding direction, left-to-right and right-to-left, from the same scanning session) were taken from each participant. No further pre-processing was performed on the data. Since we were not focused on across-participant or within-participant variability, and for computational efficiency, we focused only on two scans from a subset of the whole Human Connectome Project dataset.

Group averaged data from 14 macaque monkeys (two female) was used from the Newcastle cohort. Surface geodesic distance and homologous regions to the human data were taken from ref. 39.

The vertex-wise map of cortical myelin was the group-average map taken from the Human Connectome Project 900-subject release; it is released in the Conte32k surface space and reduced to the same 10,000 vertices as the fMRI data. Similarly, the Yeo cortical parcellation[4] in Conte32k surface space was taken from the same HCP 900 data release and was also reduced to 10,000 vertices. The 50 Neurosynth data-derived topic maps were downloaded in MNI152 2 mm space and then projected onto the mid-thickness Conte32k surface using the Connectome Workbench[40] and then reduced to the same 10,000 vertices. Topics that were not related to cognitive tasks/states were removed, leaving 24 topics.

### Geodesic distance

Pairwise geodesic distance was calculated along the cortical surface between all vertices (excluding the medial wall) using the Connectome Workbench tools, as implemented through the BrainSmash toolbox[41]. This was done on each hemisphere's mid-thickness Conte32k surface reduced to 10,000 vertices prior to calculating the distances. The resulting vertex-wise distance matrices were used in all subsequent analyses.

### Functional connectivity

The functional connectivity affinity matrix was first calculated between all 10,000 vertices for each individual fMRI scan using Pearson's correlation between the BOLD time series. For group-average results, the correlation coefficients were subsequently Fisher transformed and then for each vertex, averaged across subjects before applying an inverse Fisher transform, resulting in values between −1 and 1 for each edge of the functional connectivity matrix. Using a bounded similarity metric (0 = no similarity, 1/−1 identical) aids comparison across individuals/vertices and facilitates interpretation for the resulting empirical variograms.

### Empirical variograms

The empirical variogram was calculated by quantifying how functional connectivity decreases in similarity as distance increases. To do this, all distances between pairs of vertices were collapsed into 20 equally spaced bins. Subsequently, dissimilarity matrix was created from the functional connectivity (1- Pearson's correlation coefficient) between pairs of vertices. These values were formed into equally spaced bins using a Gaussian smoothing function (following the approach set out in refs. 41,42). This resulted in a whole-cortex empirical variogram. For vertex-wise variograms, the same approach was taken but repeated for every row of the functional connectivity/distance matrix separately, resulting in a simplified form of the empirical variogram for each vertex. The empirical variogram captures the rate of change of (dis)similarity along the cortical surface, either globally or locally for each vertex.

### Theoretical variogram

It is common practice to fit a function to empirical variograms, this is typically used prior to spatial regression; however, in our case, it allows us to compactly summarise the shape of the empirical variogram with a small number of parameters, facilitating comparisons across datasets and vertices, and aggregation across multiple vertices. For the reported analyses we used an exponential function. This is motivated by a range of prior studies suggesting exponential relationships between distance and various neural measures (e.g., ref. 43). We also performed a similar fit for three other theoretical models (a Gaussian, a power-law model, and a periodic model which allows for non-monotonic functions), with qualitatively similar results. Empirical variograms were trimmed to bins between 2 and 19 (to remove bins with few sampled distances). Subsequently, non-linear least squares was used to

estimate the sill and range parameters. Given that the distribution of pairwise distances varies across vertices (see Supplementary Fig. 1, left), for the main analyses we restricted the maximum distance to be 150 mm for calculating bins. However, qualitatively similar results were obtained by varying the upper distance limit (see Supplementary Fig. 1, right).

### Low-dimensional embedding of functional connectivity

The principal connectivity gradient was calculated using the Brain-space toolbox[44]. This involved taking the group-average functional connectivity affinity matrix and performing non-linear dimensionality reduction using the Laplacian Eigenmaps approach, separately for each hemisphere.

### Clustering

Agglomerative hierarchical clustering, with ward linkage and the Euclidean distance metric was applied simultaneously to all the vertex-wise variograms separately for each cortical hemisphere. Subsequently, SciPy's *fcluster* command was used to flatten the hierarchy into two clusters. To assess the robustness of the resulting clusters each vertex's variogram was correlated with all other variograms calculated in a separate fMRI run within the same individual. The correlation scores were Fisher transformed and then subsequently averaged both within and across clusters.

### Cognitive tasks

From the Neurosynth 50 data-derived topics dataset[22], those that did not refer to cognitive or behavioural states were removed, leaving: cognitive, inhibition, motor, numerical, action, conflict, spatial, emotion, empathy, decision, pain, memory, language, semantic, face, imagery, visual, eye movement, motion, attention, auditory, reward, social and working memory. The corresponding map for each topic was thresholded (absolute value $z > 10$, although qualitatively similar results were observed for other thresholds) and binarized, resulting in a vertex-wise mask of values that were strongly implicated for that topic (other thresholds produced qualitatively similar results). For each topic, the range and sill (taken from the theoretical variogram from the group average functional connectivity analysis) for each vertex within each mask were averaged together.

### Myelin

The estimated intracortical myelin maps derived from the ratio of T1 and T2 weighted MR images[24] from the Human Connectome Project were split into deciles based on their estimated myelin level. The empirical variograms of vertices within each decile were averaged. In addition, the estimated average myelin value for each of the clusters (see above) were calculated.

### Null models

We used spin permutation tests to assess the strength of correlations between theoretical variogram parameters with the principal gradient and estimated myelin spatial maps. A 1000 permutations of randomly rotated data were generated for the spatial maps using ref. 45 and permutation correlation values were then compared to the true value, resulting in a *p*-value. We also applied a similar approach to spinning the Yeo7 parcellation on the sphere 1000 times, and then calculating the difference in estimated range and sill between each of the Yeo7 networks; this resulted in a distribution of random differences for the sill and the parameter against which the true difference scores could be assessed.

We also used generative null models (Supplementary Fig. 2)[46] both to generate alternate statistics but also to illustrate the difference between homogeneous spatial dependency structure and the observed heterogeneous structure. To this end, three generative null models were applied to a downsampled (for computational efficiency) version

of the empirical functional connectivity matrix from which the variograms were generated: full random permutation, Mantel permutation (that preserved row and column structure), or Mantel permutation followed by spatial smoothing matched to the empirical variogram and then resampling (similar to the approach taken by[41] but applied to the functional connectivity matrix). All three approaches enforce an approximately homogeneous spatial dependency across the brain, although in the case of the randomisation and Mantel randomisation the spatial dependency is destroyed. For approach 3, a smoothing kernel was chosen iteratively to maximise the overlap with the measured empirical variogram; thereby approximately capturing the whole-brain spatial dependency but with an homogeneous spatial relationship. Each model was recalculated 1000 times, and the sill and the range for each vertex calculated. The true sill and range parameters could then be compared to the equivalent null model parameters.

### Reporting summary

Further information on research design is available in the Nature Portfolio Reporting Summary linked to this article.

## Data availability

The data used in this study are available from the Human Connectome Project (https://www.humanconnectome.org/study/hcp-young-adult/document/extensively-processed-fmri-data-documentation), the PRI-MatE Data and Resource Exchange (https://fcon-1000.projects.nitrc.org/indi/indiPRIME.html) and Neurosynth https://neurosynth.org/analyses/topics/). Data to recreate the variograms in Fig. 3 is available in source data file. Source data are provided with this paper.

## Code availability

Python code to reproduce the analyses and figures is available at https://github.com/ActiveNeuroImaging/BrainVariograms.git.

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

## Acknowledgements

R.L. was funded by the Medical Research Council (Ref: MR/R005370/1), Wellcome/EPSRC Centre for Medical Engineering (Ref: WT 203148/Z/16/Z), Simons Foundation (SFG640710) and support from the Data to Early Diagnosis and Precision Medicine Industrial Strategy Challenge Fund, UK Research and Innovation (UKRI). The authors would also like to acknowledge support from the Data to Early Diagnosis and Precision Medicine Industrial Strategy Challenge Fund, UK Research and Innovation (UKRI).

## Author contributions

R.L. and J.S. designed the study; R.L., R.W.V.D.W., F.V., T.X., R.A.B., R.S., R.M.B., M.P.M., J.R., B.C.B., E.J.H.J., E.J., D.S.M. and J.S. contributed to the analysis of the data and the manuscript.

## Competing interests

The authors declare no competing interests.
