## [Peer Review File—NEW · Nature Communications]

Variation in spatial dependencies across the cortical mantle discriminates the functional behaviour of primary and association cortexReviewer #1 (Remarks to the Author):

Variation in spatial dependencies across the cortical mantle discriminates the functional behaviour of primary and association cortex

Robert Leech, Reinder Vos De Wael, Frantisek Vasa, Ting Xu, R. Austin Benn, Robert Scholz, Rodrigo M. Braga, Michael Milham, Jessica Royer, Boris Bernhardt, Emily Jones, Elizabeth Jefferies, Daniel Margulies and Jonathan Smallwood

Summary

This paper investigates autocorrelation patterns across the cortex quantified using variograms to establish that functional similarity changes gradually within early sensorimotor cortices and rapidly in association cortices. Further analyses show that this pattern was linked to myelination and task-based systems.

General

Gaining insights into the role of brain spatial organization is an important direction of research. However, despite the use of a novel approach (variograms) it is unclear what new insights this paper offers and there is a lack of rigorous statistical/null tests to demonstrate the added value of the new approach. Please find detailed comments below, approximately in order of importance.

Major comments

1. The novelty of the results appears limited. What have we learned from the variograms that we didn't already know from gradient, ICA, ReHo and other existing methods? I know that the variogram cost function is different from these other approaches, but the results do not appear to add many novel insights. It seems that the variogram analysis is just sensitive to the same information in the data that drives other existing analyses. I would like to see a toy model example of the type of insight that could be obtained from variograms that is not already apparent from existing analyses. Without this, variograms add yet another rsfMRI analysis into the mix without a clear need, interpretational benefit, or use-case. The limited novelty is apparent in some of the write-up, for example in the discussion it is stated that '... the spatial dependence profile recapitulate the distinction between primary sensorimotor and transmodal association cortex'.
2. The results section is relatively descriptive in nature and lacks statistical comparisons or null models. This is somewhat linked to the previous point in that it is unclear what hypothesis/interpretation is specifically being tested. For example, the similarity with gradient results seems expected and might occur in non-brain data (as long as there is some basic smoothness to establish autocorrelation). I would encourage the authors to move beyond descriptive and towards hypothesis-based tests.
3. The binning of empirical variograms appears somewhat arbitrary and perhaps unnecessary. Please explain why this step is needed and test whether the results are stable across different bin choices.
4. The 'limbic network' in Fig 3 is known to suffer from drop-out and other signal issues, especially in the HCP dataset. I would be cautious to not over-interpret this as the highest sill network.
5. Misalignment is potentially an issue for vertex-wise variograms that are averaged across participants. Please clarify whether data aligned using MSM-all were used. MSM-all is the best available alignment approach, see Coalson, T. S., Van Essen, D. C., & Glasser, M. F. (2018). The impact of traditional neuroimaging methods on the spatial localization of cortical areas. *Proceedings of the National Academy of Sciences of the United States of America*, 115(27), E6356–E6365. <https://doi.org/10.1073/pnas.1801582115>
6. Macaque data are briefly included in Fig 3, but it is unclear to me what these data add and what insights are gained by including these data. I would recommend either removing these results or clarifying their added insight.
7. The link to python code in the pdf did not work, and the uploaded zip file with code contained HCP data which is in violation with the HCP data use policies. Please make sure to share code (and NO data) publicly via Git or OSF.

Minor comments:

- A. It isn't quite clear which HCP data were used. The results state '51 participants... from whom

there are two sessions separated by approximately six months', whereas the methods state 'the first 51 participants'. I thought there were only 40-ish proper test-retest people in the HCP? Please clarify which exact sample was used and if the sample included any twin/familial structure?

B. Why were only 2 resting state runs used, instead of the full 4 (or perhaps 8 if the data are test-retest participants) available runs?

C. The paper states: "To this end, we averaged vertex-wise estimates of the range and sill parameters for responsive vertices (defined as those with an estimated evoked BOLD response greater than the threshold)." Please clarify what the threshold is and what analyses are performed to estimate the evoked BOLD responses.

D. The description of the neurosynth analyses is unclear. For example, I'm not sure what the '50 Neurosynth data derived topic maps' refer to.

E. Please label figure elements A, B, C etc. in Fig 2 (and elsewhere) and refer to these in the legend. At the moment some of the legends are hard to follow and connect to the correct figures.

F. Fig. 4 is missing axis labels for most of the component graphs.

Reviewer #2 (Remarks to the Author):

This manuscript describes an approach to capturing the association between topographical distance and functional dissimilarity across the cortex. The authors evaluated two indices, the sill and range, for estimation of the functional distinction and spatial dependency, respectively. They found a distinction in the two indices between the primary sensorimotor and transmodal association cortices. Moreover, they demonstrated that the spatial distributions of the sills and ranges also corresponded to the maps of the principal gradient, cortical myelin, and represented different cognitive states.

Overall, the authors aimed to quantify the impact of spatial organization on brain functioning with spatial variogram, which was demonstrated as a reliable and sensitive measurement. However, my major concern is that the idea of associating the functional similarity with spatial topography has been intensively studied and it is not clear whether the perspective from spatial variogram diverges enough from previously published work to be considered novel. Another criticism regarding the work is the rationality of describing brain functioning with a unified index of global dissimilarity, especially considering the parallel, distributed pattern of higher-order functions. The detailed considerations are outlined below.

Major concerns:

Regarding the novelty issue, the spatial dependency of similarity in structural and functional connectivity has been demonstrated by many previous studies (e.g., Ercsey-Ravasz et al., 2013, *Neuron*; Mišić et al., 2014, *PLoS One*; Song et al., 2014, *PNAS*; Choi & Mihalas, 2019, *PLoS computational biology*). The correlation between topography and function is also a premise for the field of functional parcellation. Moreover, the cited paper of Oligschläger (2017) has already revealed the distinct geodesic distances from the primary regions and association cortices to their functionally connected regions. Such measurement that considers both spatial distance and functional similarity is quite similar to the index of variogram. The authors should clarify how their findings add onto the existing literatures and deepen the understanding of the impact of topography on brain functioning in a distinct way.

Second, regarding the variogram framework, the spatial profiles derived from averaging dissimilarity along with distance can be methodologically biased for multimodal regions, as the method undermines the role of long-range connections and distributed pattern of higher-order cognition. I also find the network-level results to be doubtful as the possible influence of the network size, which affects the range of the embedded vertices, on the variogram was not precluded. Considering these points, I would recommend the authors to clarify whether and how the spatial variogram can be suitable for depicting the spatial characteristics of association

networks.

Furthermore, additional quantitative analyses are required to support the authors' conclusions. It would be more convincing to explain the relationship between the sills and ranges, the result of statistic comparison of the sill and range across networks, and the spatial similarity between the myelin/gradient map and sill/range map. Moreover, the indices chosen to validate the role of variogram (i.e., principal gradient, cortical myelin, and maps of cognitive states) are themselves spatially homogenous and uniformly reflect the unimodal-multimodal alteration. It is not surprising to yield consistent results by associating those indices. It would be helpful to clarify why these results are not redundant, or add more independent/heterogeneous indices to substantiate their findings.

Minor concerns or suggestions:

1. The authors derived two indices, the sill and range, from the variogram. However, the authors did not attempt to interpret the distinguished roles of sill and range. It remains hard for me to grasp the distinct contribution of the two indices. The authors may also consider discussing potential physiological implications of the sill and range indices.
2. It seems that the association between the sill and range is not as simple as a reverse relationship (Figure 2). However, the authors claimed that the multimodal regions showed high sills and short effective ranges, whereas the unimodal regions demonstrated the opposite. Such argument lacks direct quantitative comparisons. Moreover, the variograms of the sensory networks show that the global functional dissimilarities continuously increase as geodesic distance, especially in macaques (Figure 3), making it hard to believe the sill derived from the asymptote within sensory regions to be lower than the multimodal regions.
3. Considering the different locations and ranges of spatial distance across the vertices, the decision of including only the vertex-pairs within 150mm seems somehow arbitrary. Did the authors explore the variogram that considering a wider or full range of distance, both for the entire cortex and each network? Would those be consistent with the existing ones?
4. The conclusion that the vertices within <40 mm have similar temporal profiles appears to be unsubstantial, as the inflection point at 30-40mm could only reveal the spatial dependency exist within such distance. I would suggest a more conservative interpretation regarding this finding.
5. The authors proposed the gradual change of variogram as mesoscale features located within sensory cortex. However, the estimated ranges in the visual and motor regions can be as long as the global distance (i.e., 150mm), indicating a macroscale gradient for the functional organization of sensory cortex. How would the authors reconcile this seemingly contradictory finding?
6. The choice of fitting an exponential function as a winning model compared with sinusoidal and gaussian may not be convincing. A relevant work from Choi & Mihalas (2019) has otherwise investigated that the spatial dependency of connectivity using a power law. The authors may also try other functions such as power or logarithm to confirm whether indices from other functions would consistently reflect the differences between unimodal and multimodal regions.
7. For the vertex-wise map of the sill values, it is hard to distinguish different brain regions with this blue-to-green colorbar. I would suggest a colorbar with more complementary colors to make brain regions with different value more distinguishable.
8. The highly similar pattern between the variogram clustering and myelin map (Figure 4) appears to be the result of selective representation of the lateral section of myelin map. As the medial view of myelin map in Glasser and Essen (2011) seems to be quite different from the variogram clustering. I would suggest the authors to add statistical evidence for the similarity between the maps.

Reviewer #3 (Remarks to the Author):

In this work, Leech et al. use variograms to investigate how functional (dis)similarity changes as a function of the distance between regions. They find that function changes gradually within sensory and motor cortex as the distance between regions increases, while in association cortex function changes rapidly over shorter distances. They also report that similar variograms profiles are present in monkeys, and that these differential classes of spatial dependency are related to variation in intracortical myelin between sensory motor and association cortex.

The authors address a topic that is of broad interest to neuroscience. The manuscript is well written and the analyses are clearly explained. I think this work makes an interesting contribution. I have few suggestions to improve this manuscript to increase its clarity and interest to the relevant communities.

Major comments:

- The first comment is more "conceptual". The authors state: "..all distances between pairs of vertices were collapsed into 20 equally spaced bins. Subsequently, the difference in functional connectivity (Pearson's correlation coefficient) between pairs of vertices was calculated and formed into equally spaced bins..".

From what I understand, what the authors define as dissimilarity (y-axis of the variograms) is the delta (the edgewise difference?) of the functional connectivity between pairs of vertices. However, the functional connectivity per se already measures the similarity/dissimilarity of two vertices' activity (in terms of Pearson's correlation). What the authors are reporting is then the rate of change in similarity across the cortex, which (to me at least) is a deeper measurement than just comparing vertex distance to FC edge connectivity directly. Could you please clarify this idea to me?

If my assumption is correct, I would then encourage the authors to elaborate more on this (important) subtlety in the Methods section, and reevaluate some claims in the main text.

- For instance, following up on the comment above, what do the authors mean by "second order non stationarity"? Are there some papers they are referring to? Rather, I believe they are measuring the empirical derivative of the covariance matrix of a (most likely) non-stationary system.

- My second comment relates to the implementation of a null model for the variograms reported in Figure 2 and Figure 3. What would be the shape of the variogram if we randomly shuffled the edges of functional connectivity matrix reported in Fig. 1 (while keeping the original geodesic matrix)? Showing that the model will fail (as I think it will) after random permutation of the functional edges will strengthen the claims of the manuscript.

Minor comments:

- Why did the authors performed the analysis on 51 subjects and not on the usual 100 HCP subjects? Please motivate the choice

- Please add label on the y-axis in Figure 2, top panel (Dissimilarity, I suppose)

- The authors state: "all distances between pairs of vertices were collapsed into 20 equally spaced bins". Why 20 bins? Do they expect the variograms to change with bin size?

- Double-check for small typos/changes in font/color in the draft

Point-by-point response to reviewers' comments

Reviewer #1

General

Gaining insights into the role of brain spatial organization is an important direction of research. However, despite the use of a novel approach (variograms) it is unclear what new insights this paper offers and there is a lack of rigorous statistical/null tests to demonstrate the added value of the new approach. Please find detailed comments below, approximately in order of importance.

Major comments

1. The novelty of the results appears limited. What have we learned from the variograms that we didn't already know from gradient, ICA, ReHo and other existing methods? I know that the variogram cost function is different from these other approaches, but the results do not appear to add many novel insights. It seems that the variogram analysis is just sensitive to the same information in the data that drives other existing analyses. I would like to see a toy model example of the type of insight that could be obtained from variograms that is not already apparent from existing analyses. Without this, variograms add yet another rsfMRI analysis into the mix without a clear need, interpretational benefit, or use-case. The limited novelty is apparent in some of the write-up, for example in the discussion it is stated that '... the spatial dependence profile recapitulate the distinction between primary sensorimotor and transmodal association cortex.'

We thank the reviewer for their comment. We have revised the manuscript based on their comments and in particular have performed several sets of additional analyses to highlight why the variogram perspective provides a different sort of insight to that from other techniques in the literature (page 7, from line 25, page 18, from line 3, and Supplementary Figure 2). Fundamentally, our study is concerned with the relationship between spatial distance and similarity of functional connectivity (see page 12, from line 34), *it is not a parcellation approach*. Although other techniques (like ICA) are likely dependent on how spatial variation impacts brain function, the variogram approach is specifically concerned with understanding how spatial variation underlies the patterns that are seen in such approaches. This logic is highlighted by the reviewer in that we are interested in understanding which features of existing parcellations are captured by variograms, and which are not. Critically, while the variograms highlight differences between primary systems and association cortex, they do not separate networks within these cortical types. Thus, one fundamental contribution of our study is that it highlights processes that contribute to gradient descriptions (i.e. Margulies et al., 2016) but not the network level (Yeo, Krienen et al., 2011). In the text of this revision, we have attempted to make this more explicit (in the introduction, page 4, from line 34; in the discussion, page 12, from line 34).

2. The results section is relatively descriptive in nature and lacks statistical comparisons or null models. This is somewhat linked to the previous point in that it is unclear what hypothesis/interpretation is specifically being tested. For example, the similarity with gradient results seems expected and might occur in non-brain data (as long as there is some basic smoothness to establish autocorrelation). I would encourage the authors to move beyond descriptive and towards hypothesis-based tests.

In the revised manuscript, we use a generative null model to illustrate the distinction between homogeneous and heterogeneous spatial relationships (see page 7, from line 25; page 18, from line 3; and Supplementary Figure 2). We shuffle and then smooth the functional connectivity matrix with a single exponential function (approximately matching the observed whole-brain variogram) and demonstrate that the resulting vertex-wise variograms are much more similar than we observe with the empirical data and do not correlate with the gradient organization. This illustrates that spatial autocorrelation on its own is not sufficient to explain relationships with other spatial maps e.g., gradients or intra-cortical myelin, but that we must also account for heterogeneous spatial relationships.

We have added several sections to the methods and the results performing null hypothesis testing using both a generative null model based on a homogeneous spatial relationship with functional connectivity, as well as spin-based permutation testing (see Figures 2, 4 and 6; page 9, from line 9; page 17 from line 43). We show that relationships

between parameters of the theoretical variograms, and intracortical myelin as well as the principal gradient, cannot simply be accounted for by a homogenous spatial relationship across cortex. Similarly, several of the Yeo7 canonical resting networks exhibit significantly different average sills and average ranges than the null model; therefore, these variogram features again occur over and above the background influence of a spatially homogenous relationship across cortex.

3. *The binning of empirical variograms appears somewhat arbitrary and perhaps unnecessary. Please explain why this step is needed and test whether the results are stable across different bin choices.*

The use of binning is to facilitate comparison of the parameters from the theoretical variograms across different vertices with different distance distributions. In the revised manuscript, we now include information about the distribution of distances and the maximum distances for different vertices which highlight that there is considerable variability in the maximum distances which could potentially lead to differences in the model fit for different regions. We also show result of the relationship between sill and range and the principal functional gradient for different upper maximum distances (and consequently wider bins), with qualitatively similar results (see Supplementary Figure 1; in the results, page 6, from line 13; in the methods, page 17, from line 1).

4. *The 'limbic network' in Fig 3 is known to suffer from drop-out and other signal issues, especially in the HCP dataset. I would be cautious to not over-interpret this as the highest sill network.*

We agree with the reviewer and have added text to the manuscript to note this point, and caution interpretation of the 'limbic network' (see page 10, from line 15).

5. *Misalignment is potentially an issue for vertex-wise variograms that are averaged across participants. Please clarify whether data aligned using MSM-all were used. MSM-all is the best available alignment approach, see Coalson, T. S., Van Essen, D. C., & Glasser, M. F. (2018). The impact of traditional neuroimaging methods on the spatial localization of cortical areas. *Proceedings of the National Academy of Sciences of the United States of America*, 115(27), E6356–E6365. <https://doi.org/10.1073/pnas.1801582115>*

We can confirm that the data were aligned using the MSM-all approach. In addition, we performed some of the analyses (e.g., Figures 1 and 5) on individual participant's data which would limit potential blurring issues when calculating average variograms for each vertex.

6. *Macaque data are briefly included in Fig 3, but it is unclear to me what these data add and what insights are gained by including these data. I would recommend either removing these results or clarifying their added insight.*

We have included clarification and greater justification of the inclusion of the macaque data in the revised manuscript in the results and the discussion section (see page 10, from line 8).

7. *The link to python code in the pdf did not work, and the uploaded zip file with code contained HCP data which is in violation with the HCP data use policies. Please make sure to share code (and NO data) publicly via Git or OSF.*

We apologize for this confusion on our part. The revised manuscript contains a link to the public Git which contains the code but no data (see link on page 18, line 22).

Minor comments:

A. *It isn't quite clear which HCP data were used. The results state '51 participants... from whom there are two sessions separated by approximately six months', whereas the methods state 'the first 51 participants'. I thought there were only 40-ish proper test-retest people in the HCP? Please clarify which exact sample was used and if the sample included any twin/familial structure?*

We apologize for the confusion here, and thank the reviewer for highlighting this. This was a mistake on our part when writing the results/methods. The data are the first 51 participants from the 1200 Subject Young Adult HCP dataset. The scans were from a single session. See page 15, from line 37 for the updated text).

B. Why were only 2 resting state runs used, instead of the full 4 (or perhaps 8 if the data are test-retest participants) available runs?

For computational reasons, we restricted ourselves to a subset of the dataset. Further, since we were not focused on across-participant variability, we focused on a small set of the data. We have clarified this in the revised manuscript (See page 15, line 40).

C. The paper states: "To this end, we averaged vertex-wise estimates of the range and sill parameters for responsive vertices (defined as those with an estimated evoked BOLD response greater than the threshold)." Please clarify what the threshold is and what analyses are performed to estimate the evoked BOLD responses.

The threshold was arbitrarily chosen to be 10, based on visual inspection of the topic maps from Neurosynth. This has now been added to the methods section (see page 17, from line 27).

D. The description of the neurosynth analyses is unclear. For example, I'm not sure what the '50 Neurosynth data derived topic maps' refer to.

We have clarified the description of the Neurosynth analysis in the revised manuscript (see page 11, from line 12).

E. Please label figure elements A, B, C etc. in Fig 2 (and elsewhere) and refer to these in the legend. At the moment some of the legends are hard to follow and connect to the correct figures.

F. Fig. 4 is missing axis labels for most of the component graphs.

We apologize for this and have corrected these mistakes.

"Reviewer #2 (Remarks to the Author):

This manuscript describes an approach to capturing the association between topographical distance and functional dissimilarity across the cortex. The authors evaluated two indices, the sill and range, for estimation of the functional distinction and spatial dependency, respectively. They found a distinction in the two indices between the primary sensorimotor and transmodal association cortices. Moreover, they demonstrated that the spatial distributions of the sills and ranges also corresponded to the maps of the principal gradient, cortical myelin, and represented different cognitive states.

Overall, the authors aimed to quantify the impact of spatial organization on brain functioning with spatial variogram, which was demonstrated as a reliable and sensitive measurement. However, my major concern is that the idea of associating the functional similarity with spatial topography has been intensively studied and it is not clear whether the perspective from spatial variogram diverges enough from previously published work to be considered novel. Another criticism regarding the work is the rationality of describing brain functioning with a unified index of global dissimilarity, especially considering the parallel, distributed pattern of higher-order functions. The detailed considerations are outlined below.

Major concerns:

Regarding the novelty issue, the spatial dependency of similarity in structural and functional connectivity has been demonstrated by many previous studies (e.g., Ercsey-Ravasz et al., 2013, Neuron; Mišić et al., 2014, PLoS One; Song et al., 2014, PNAS; Choi & Mihalas, 2019, PLoS computational biology). The correlation between topography and function is also a premise for the field of functional parcellation. Moreover, the cited paper of Oligschläger (2017) has already revealed the distinct geodesic distances from the primary regions and association cortices to their functionally connected regions. Such measurement that considers both spatial distance and functional similarity is quite similar to the index of variogram. The authors should clarify how their findings add onto the existing literatures and deepen the understanding of the impact of topography on brain functioning in a distinct way."

We thank the reviewer for their insightful comments. While the existence of spatial dependencies has been long recognised in functional and structural neuroimaging, it is almost exclusively considered as a unitary or homogeneous phenomenon. For example, Ercsey-Ravasz et al, 2013 look for a single relationship between space and connectivity, resulting in a single rule quantifying spatial dependency in terms of a single parameter for the exponential function. Similarly, Song et al, and Mišić et al quantify single cortex-wide relationships between connectivity and distance. These results are comparable to the whole-brain variograms used in prior work on generative null models for spatial autocorrelation (e.g. Burt et al. Nat. Neurosci. 2018, NeuroImage 2020) that we also present in Figure 1 and 2. In contrast, in our present work, we assess whether individual vertices along the cortical surface have different distance-dependent relationships. We further show that this regional heterogeneity relates to well recognized variability observed in prior large-scale functional and structural analyses and so may serve important organizing and functional roles. Choi & Milhalas do note that a single power-law relationship is inadequate at capturing neural dynamics unless additional constraints are incorporated. This is consistent with our observations of heterogeneity of the distance-dependent relationships across the cortex; however, they still frame their results in terms of a single spatial dependency across the brain. We have reworked the text particularly the discussion to consider these prior works in more detail and situate what our work adds.

Furthermore, even though other authors have considered whether spatial dependencies are invariant our study establishes which features of brain organisation are impacted by varying impact of space on function. For example, it is clear that broad features of brain organisation (like those captured by the principal gradient) are associated with variation in spatial dependencies, however, fine grained features of association cortex (for example the distinction between networks in this type of cortex) are not. In this revision we have attempted to make this novel contribution of our work more explicit (see page 7, from line 25; page 12, from line 34; and Supplementary Figure 2)

Second, regarding the variogram framework, the spatial profiles derived from averaging dissimilarity along with distance can be methodologically biased for multimodal regions, as the method undermines the role of long-range connections and distributed pattern of higher-order cognition.

We agree with the reviewer that the distribution of long-range connections and parallel, distributed functional organization within some brain regions will substantially change its distance dependent relationship measured with vertexwise variograms. We do not see this as a methodologically-biased approach, however, since we are using it to describe how function in one region relates to the cortical environment in which it is embedded. It is likely that regions receiving long distance communication may be more locally heterogeneous than those which do not. However, since the primary goal of our paper is to assess whether spatial similarity is homogeneous across the cortex, the disturbances to the local homogeneity that long distance communication causes cannot undermine our argument that spatial variation is heterogeneous – it is instead a mechanism that may account for the changes in homogeneity. In this revision we have made this point explicit, and explicitly considered the reviewers suggested mechanism in the discussion (see page 15, from line 14).

I also find the network-level results to be doubtful as the possible influence of the network size, which affects the range of the embedded vertices, on the variogram was not precluded. Considering these points, I would recommend the authors to clarify whether and how the spatial variogram can be suitable for depicting the spatial characteristics of association networks.

We have performed null hypothesis testing with spin permutations of the Yeo7 networks on the surface. Importantly, the rotated versions of the Yeo networks have the same number of included vertices and same spatial shape, but their spatial location is different. These results show that several canonical resting-state networks significantly differ from each other in terms of their sills and/or ranges relative to their spatially permuted counterparts; in particular, networks with from different types of cortex (sensorimotor versus higher-order cortex show significant differences from each other, but not within types of cortex. See Figures 4 and 6; page 9, from line 9; and page 17 from line 43.

Furthermore, additional quantitative analyses are required to support the authors' conclusions. It would be more convincing to explain the relationship between the sills and ranges, the result of statistic comparison of the sill and range across networks, and the spatial similarity between the myelin/gradient map and sill/range map.

We have performed substantial additional analyses to statistically test the relationships using spin permutation testing and show that the relationships are statistically robust for the myelin/gradient and sill/range analyses. We have also conducted spin-based analyses to identify statistical differences between Yeo networks (see above). See Figures 2, 4 and 6; page 9, from line 9; and page 17 from line 43.

Moreover, the indices chosen to validate the role of variogram (i.e., principal gradient, cortical myelin, and maps of cognitive states) are themselves spatially homogenous and uniformly reflect the unimodal-multimodal alteration. It is not surprising to yield consistent results by associating those indices. It would be helpful to clarify why these results are not redundant, or add more independent/heterogeneous indices to substantiate their findings.

We agree. The purpose of our analysis is to understand which of the existing methods of describing brain function are spatially homogeneous. It is clear from our analysis that some are (the ones the reviewers describe) but others are not (e.g., the Yeo Networks). Thus, the value of our paper is to formally establish (for example) that the principal gradient is a result of spatial heterogeneity but the Yeo networks are not, even though, as has been previously established, the principal gradient organises the Yeo Networks (Margulies et al., 2016, PNAS). We briefly consider this in the revised manuscript (see page 4 from line 34; page 9, from line 9; page 14, from line 40).

Minor concerns or suggestions:

1. The authors derived two indices, the sill and range, from the variogram. However, the authors did not attempt to interpret the distinguished roles of sill and range. It remains hard for me to grasp the distinct contribution of the two indices. The authors may also consider discussing potential physiological implications of the sill and range indices.

We have added to the introduction and the discussion to try to elaborate on the two indices. Particularly the role of the Y-axis as the correlation coefficient and what the asymptote of this implies (see page 6, from line 24).

2. It seems that the association between the sill and range is not as simple as a reverse relationship (Figure 2). However, the authors claimed that the multimodal regions showed high stills and short effective ranges, whereas the unimodal regions demonstrated the opposite. Such argument lacks direct quantitative comparisons. Moreover, the variograms of the sensory networks show that the global functional dissimilarities continuously increase as geodesic distance, especially in macaques (Figure 3), making it hard to believe the sill derived from the asymptote within sensory regions to be lower than the multimodal regions.

We have added additional analyses (as detailed above) to statistically quantify the relationship of the sill and range with the Principal Gradient (Figure 2, bottom right and page 7, from line 15) and the individual Yeo network (page, 9, from line 9 and Figure 4). The analyses highlight that the sill and range do vary in opposite directions with regard to the unimodal-multimodal split, but that the relationship is more complex than the sill and range being simply anti-correlated.

3. Considering the different locations and ranges of spatial distance across the vertices, the decision of including only the vertex-pairs within 150mm seems somehow arbitrary. Did the authors explore the variogram that considering a wider or full range of distance, both for the entire cortex and each network? Would those be consistent with the existing ones?

We now include results for a wider range of variogram distances (see Supplementary Figure 1 and page 6, from line 13). These do not qualitatively change the relationships we observed. We also include the distribution of distances across the cortex and the distribution of maximum distances for each vertex (see Supplementary Figure 1).

4. The conclusion that the vertices within <40 mm have similar temporal profiles appears to be unsubstantial, as the inflection point at 30-40mm could only reveal the spatial dependency exist within such distance. I would suggest a more conservative interpretation regarding this finding.

We agree with the reviewer and have taken a more conservative interpretation in the revised manuscript, removing this point (see page 6, from line 13 for the revised text).

5. *The authors proposed the gradual change of variogram as mesoscale features located within sensory cortex. However, the estimated ranges in the visual and motor regions can be as long as the global distance (i.e., 150mm), indicating a macroscale gradient for the functional organization of sensory cortex. How would the authors reconcile this seemingly contradictory finding?*

We have revised the discussion and the need for future work in light of this comment (page 15, from line 18).

6. *The choice of fitting an exponential function as a winning model compared with sinusoidal and gaussian may not be convincing. A relevant work from Choi & Mihalas (2019) has otherwise investigated that the spatial dependency of connectivity using a power law. The authors may also try other functions such as power or logarithm to confirm whether indices from other functions would consistently reflect the differences between unimodal and multimodal regions.*

We have recalculated the results including a power law relationship as well as the others we considered previously. The Power Law fit performs, in general, worse than the exponential relationship. We note that differences in the literature may relate, in part, to aspects of fMRI and BOLD signal and may not be a general relationship across different brain modalities and species (as with Choi & Mihalas). We have amended the manuscript in the methods and results to include the power-law relationship. (see page 6, line 20; page 16, line 47).

7. *For the vertex-wise map of the sill values, it is hard to distinguish different brain regions with this blue-to-green colorbar. I would suggest a colorbar with more complementary colors to make brain regions with different value more distinguishable.*

We have revised the figure with a different colormap (see Figure 2).

8. *The highly similar pattern between the variogram clustering and myelin map (Figure 4) appears to be the result of selective representation of the lateral section of myelin map. As the medial view of myelin map in Glasser and Essen (2011) seems to be quite different from the variogram clustering. I would suggest the authors to add statistical evidence for the similarity between the maps."*

We have added statistical analyses based on spin permutation testing, that show a significant relationship. We also now show (in the Figure 6) the (nominal) correlation value (approximately 0.5 for the range and -0.4 for the sill) between myelin and the variogram parameters and present both medial and lateral views of the HCP myelin map.

Reviewer #3 (Remarks to the Author):

In this work, Leech et al. use variograms to investigate how functional (dis)similarity changes as a function of the distance between regions. They find that function changes gradually within sensory and motor cortex as the distance between regions increases, while in association cortex function changes rapidly over shorter distances. They also report that similar variograms profiles are present in monkeys, and that these differential classes of spatial dependency are related to variation in intracortical myelin between sensory motor and association cortex.

The authors address a topic that is of broad interest to neuroscience. The manuscript is well written and the analyses are clearly explained. I think this work makes an interesting contribution. I have few suggestions to improve this manuscript to increase its clarity and interest to the relevant communities.

Major comments:

- *The first comment is more "conceptual". The authors state: "...all distances between pairs of vertices were collapsed into 20 equally spaced bins. Subsequently, the difference in functional connectivity (Pearson's correlation coefficient)*

between pairs of vertices was calculated and formed into equally spaced bins..".

From what I understand, what the authors define as dissimilarity (y-axis of the variograms) is the delta (the edgewise difference?) of the functional connectivity between pairs of vertices. However, the functional connectivity per se already measures the similarity/dissimilarity of two vertices' activity (in terms of Pearson's correlation). What the authors are reporting is then the rate of change in similarity across the cortex, which (to me at least) is a deeper measurement than just comparing vertex distance to FC edge connectivity directly. Could you please clarify this idea to me?

If my assumption is correct, I would then encourage the authors to elaborate more on this (important) subtlety in the Methods section, and reevaluate some claims in the main text.

We thank the reviewer for their comment. We realize that the presentation in the manuscript could have been much clearer and have changed the figure legends, and clarified the methods text. The variogram is calculated based on dissimilarity (i.e., simply 1- correlation coefficient) at different distances; in places we implied it was the difference in correlation coefficients, which is incorrect. The resulting variogram, however, is a measurement of rate of change of similarity with cortical distance; summarized by the range and the sill for the theoretical variogram approximation. We now elaborate on this more in the main text, both in the methods section and in the introduction (see Figure 1; page 5, from line 6; page 6, from line 24; page 16, from line 29).

- For instance, following up on the comment above, what do the authors mean by "second order non stationarity"? Are there some papers they are referring to? Rather, I believe they are measuring the empirical derivative of the covariance matrix of a (most likely) non-stationary system.

We again thank the reviewer for this comment; we have removed reference to second-order non-stationarity. As the reviewer pointed out, we are not actively measuring it; instead, we now make reference to regional heterogeneity in empirical/theoretical variogram models of spatial dependency (see, e.g., page 10, line 8).

- My second comment relates to the implementation of a null model for the variograms reported in Figure 2 and Figure 3. What would be the shape of the variogram if we randomly shuffled the edges of functional connectivity matrix reported in Fig. 1 (while keeping the original geodesic matrix)? Showing that the model will fail (as I think it will) after random permutation of the functional edges will strengthen the claims of the manuscript.

We have now added substantial additional null models. The statistical relationship with other neural measures (functional gradients, Yeo networks and estimated intracortical myelin have been assessed with spin-based permutation tests (Figures 2 (bottom), 5 and 6 (bottom, right)). However, more relevantly, we have also built different null models with either full random shuffling, Mantel shuffling (preserving within row/column structure) and Mantel shuffling followed by smoothing and resampling to approximately match the empirical whole-brain variogram (Supplementary Figure 2, top, page 8, from line 27; page 22, from line 3). All three of these null models assume an approximately homogeneous spatial relationship across the whole brain; the first two (non-spatial null models) show a relatively flat variogram, whereas the spatially smoothed homogeneous model shows an approximately similar whole-brain spatial variogram (top, right), but one with substantially less regional variability (Supplementary Figure 2, middle), and no relationship with the principal functional gradient (middle and bottom).

Minor comments:

- Why did the authors performed the analysis on 51 subjects and not on the usual 100 HCP subjects? Please motivate the choice

We intended to run 50 participants but an indexing error resulted in 51. The number was chosen purely for computational efficiency reasons. We have added to the methods to state this.

- Please add label on the y-axis in Figure 2, top panel (Dissimilarity, I suppose)

This has been done.

- *The authors state: "all distances between pairs of vertices were collapsed into 20 equally spaced bins". Why 20 bins? Do they expect the variograms to change with bin size?*

The bin choice and size was somewhat arbitrary. In Supplementary Figure 1 (right), we now show the results of different bin widths (by the maximum length) and demonstrate that the distribution of sills/ranges and relationship with the principal functional gradient remains approximately the same.

- *Double-check for small typos/changes in font/color in the draft*

This has been done.

Reviewer #1 (Remarks to the Author):

I thank the authors for their thorough and thoughtful revision. The clarifications and addition of null models have greatly strengthened the paper. I just have a few remaining comments:

1. The authors report that spatial autocorrelation on its own is not sufficient. It would be helpful to discuss the relationship between the manuscript's findings and this recent paper in the discussion section (<https://pubmed.ncbi.nlm.nih.gov/37339700/>).
2. In the discussion, the text states that whole brain variograms are reasonably consistent (...) within individuals measured at different timepoints. However, the two scans were acquired on the same day, immediately after each other (i.e., not 'at different timepoints'). It would be suitable to edit the text to clarify this in the discussion, legend of figure 2, methods, and results.
3. In figure 2, the added null model description refers to panel E, but it should be panel H.

Reviewer #3 (Remarks to the Author):

The authors have addressed all my concerns.

Reviewer #1

I thank the authors for their thorough and thoughtful revision. The clarifications and addition of null models have greatly strengthened the paper. I just have a few remaining comments:

1. The authors report that spatial autocorrelation on its own is not sufficient. It would be helpful to discuss the relationship between the manuscript's findings and this recent paper in the discussion section (<https://pubmed.ncbi.nlm.nih.gov/37339700/>).

We now briefly discuss this article at the very end of the discussion.

2. In the discussion, the text states that whole brain variograms are reasonably consistent (...) within individuals measured at different timepoints. However, the two scans were acquired on the same day, immediately after each other (i.e., not 'at different timepoints'). It would be suitable to edit the text to clarify this in the discussion, legend of figure 2, methods, and results.

This has been changed throughout the document (i.e., page 5, line 35; page 6, line 5; page 8, line 11; page 9, line 12; page 20, line 16).

3. In figure 2, the added null model description refers to panel E, but it should be panel H.

This has been changed.